# Influenza vaccination of school teachers: A scoping review and an impact estimation

**Anne Huiberts** [1] *, **Brigitte van Cleef**[2], **Aimée Tjon-A-Tsien**[3], **Frederika Dijkstra**[1], **Imke Schreuder**[1], **Ewout Fanoy**[3], **Arianne van Gageldonk**[1], **Wim van der Hoek**[1], **Liselotte van Asten**[1]

**1** Centre for Infectious Disease Control Netherlands, National Institute for Public Health and the Environment (RIVM), Bilthoven, The Netherlands, **2** Department for Infectious Disease Control, Public Health Service, Amsterdam, The Netherlands, **3** Department for Infectious Disease Control, Public Health Service, Rotterdam-Rijnmond, The Netherlands

* Annehuiberts@hotmail.nl

## Abstract

### Introduction

Influenza vaccination, besides protecting traditional risk groups, can protect employees and reduce illness-related absence, which is especially relevant in sectors with staff shortages. This study describes current knowledge of influenza vaccination in teachers and estimates its potential impact.

### Methods

We conducted a scoping review of the considerations for and impact of influenza vaccination of schoolteachers (grey and scientific literature up to 2020 March, complemented with interviews). We then estimated the potential impact of teacher vaccination in the Netherlands, with different scenarios of vaccine uptake for 3 influenza seasons (2016–2019). Using published data on multiple input parameters, we calculated potentially averted absenteeism notifications, averted absenteeism duration and averted doctor visits for influenza.

### Results

Only one scientific paper reported on impact; it showed lower absenteeism in vaccinated teachers, whereas more knowledge of vaccination impact was deemed crucial by 50% of interviewed experts. The impact for the Netherlands of a hypothetical 50% vaccine uptake was subsequently estimated: 74–293 potentially averted physician visits and 11,178–28,896 potentially averted days of influenza absenteeism (on ≈200,000 total teacher population). An estimated 12–32 vaccinations were required to prevent one teacher sick-leave notification, or 3.5–9.1 vaccinations to prevent one day of teacher absenteeism (2016–2019).

### Conclusion

Scientific publications on influenza vaccination in teachers are few, while public interest has increased to reduce teacher shortages. However, school boards and public health experts

**Data Availability Statement:** All data on scientific literature, grey literature, interviews, and media articles are within the paper and its Supporting information files.

**Funding:** This study was financed from the budget of the RIVM, made available by the Ministry of Health, Welfare and Sport, project number V/ 150207/20/RI. The funders had no role in study design, data collection and analysis, decision to publish, or preparation of the manuscript.

**Competing interests:** The authors have declared that no competing interests exist.

**Abbreviations:** ECDC, European Centre for Disease Prevention and Control; EU, European Union; GP, general practitioner; PHS, public health service; UK, United Kingdom; USA, United States; VE, vaccine effectiveness; WHO, World Health Organization.

indicate requiring knowledge of impact when considering this vaccination. Estimations of 3.5–9.1 vaccinated teachers preventing one day of influenza-related sick leave suggest a possible substantial vaccination impact on absenteeism. Financial incentives, more accessible on-site vaccinations at workplaces, or both, are expected to increase uptake, but more research is needed on teachers' views and vaccine uptake potential and its cost-effectiveness. Piloting free on-site influenza vaccination in several schools could provide further information on teacher participation.

## Introduction

In the Netherlands and the European Union (EU), up to 2020, influenza infections reflected the highest burden of disease of all infectious diseases in disability-adjusted life years [1]. Annually, influenza infection causes many cases of influenza-like illness in the general population, hospitalisations and intensive care admissions [1]. It also contributes substantially to seasonal mortality rates [2]. The most important preventive measure is vaccination of risk groups with season-specific influenza vaccines [3, 4]. With influenza viruses constantly evolving (antigenic drift) [1] and waning antibody levels, a new vaccination campaign is needed annually [5]. The seasonal vaccine offers partial (roughly 30–50%) protection against medically attended, symptomatic infection if the match between the vaccine viruses and the circulating viruses is strong [6]; it provides less protection if circulating strains deviate from predicted.

Traditional risk groups targeted for influenza vaccination in many countries include the elderly (60+) and persons with underlying chronic conditions, because of their higher risk of influenza complications. In the Netherlands, this does not include teachers unless they are individually targeted because of fitting one of those traditional risk groups. In some countries, such as the United Kingdom (UK), healthy children are also eligible [1, 7]. The dual objective of such eligibility is to directly protect children while indirectly decreasing transmission to risk groups such as the elderly, since infants and schoolchildren are considered the drivers of influenza transmission in the wider community [1]. Influenza viruses are easily transmitted from person to person, particularly in crowded locations such as schools. European countries focus on risk groups, whereas in the United States (USA), routine annual influenza vaccination is recommended for all persons aged ≥6 months [6]. The Netherlands, as most European countries, also recommends annual vaccination of healthcare workers [1, 8], primarily aiming to protect their vulnerable patients.

Influenza vaccination may also protect employees in sectors other than health care and might reduce absenteeism due to influenza-associated illnesses [9]. This is especially relevant in sectors with staff shortages and where an influenza epidemic can have a profound impact on business continuity [9]. In the Netherlands, the healthcare sector and the educational system, particularly, the schoolteacher workforce, are especially vulnerable to disruption due to staff shortages [10–13].

Dutch media have reported on the lack of substitute teachers that has caused problems for schools in recent years [14, 15]. In the absence of national guidance on influenza vaccination in the educational system, some local companies and organisations encourage their staff to be vaccinated. For example, in Amsterdam, the public health service has offered free vaccination to schoolteachers since 2018/2019. However, little is known about this vaccination experience, its impact on teachers, or any similar considerations and policies in other municipalities. To our knowledge, no summary of available evidence on teacher vaccination is available.

Although Dutch media have recently reported on influenza vaccination of teachers as one way to decrease schoolteacher shortages, a comprehensive overview of literature is lacking. Therefore, we describe the current knowledge of teacher influenza vaccination using very broad input (scientific literature, grey literature, Dutch newspaper reports and information from key informants). Since knowledge of the impact of this vaccination in teacher populations is crucial for decision-making, but is lacking, we then estimated its potential impact.

## Methods

We conducted a scoping review of the considerations for and impact of influenza vaccination of school teachers. We then estimated the potential impact of teacher vaccination in the Netherlands at different scenarios of vaccine uptake for three influenza seasons (2016–2019).

### Scoping review

We conducted a scoping review of influenza vaccination of schoolteachers. The scoping review was based on the framework of Arksey and O'Malley [16]. We studied grey and scientific literature published up to March 2020. The literature was charted for study characteristics, details of vaccination implementation, vaccination uptake, teachers' attitudes, and vaccination effect (impact).

Key information was extracted by one reviewer (S1 and S2 Files). This literature study was accompanied by interviews of 10 key informants in the two largest Dutch cities in 2020 (S3 File) and Dutch newspaper reports published from 2010 to 2019 (S4 File).

A scoping review was chosen, because our aim was to summarize the available evidence from both scientific and grey sources and to identify knowledge gaps in a specified research area where a comprehensive overview of literature is lacking [16, 17]. With a scoping review, we could address broader research questions and allow for redefinition of terms during the search process than we could with a systematic review. In a scoping review, authors do not typically assess the quality of the studies [17, 18], but we were able to exercise the option of including information from interviews with key informants.

### Impact calculation

For multiple hypothetical vaccine uptake scenarios (2%, 10%, 25%, 50% and 70%), the potential impact of vaccinating schoolteachers for influenza was calculated by estimating the number of events the vaccine uptake would have averted (in seasons 2016/2017 to 2018/2019). We estimated the averted numbers of three different influenza-related events:

1. The potentially averted number of influenza *absenteeism notifications* among teachers (calculation in Fig 1).

2. The potentially averted influenza *absenteeism duration* (i.e., the number of *days* of influenza absenteeism among teachers) (calculation in Fig 1).

3. The potentially averted number of *general practitioner (GP) visits* for influenza by teachers (calculation in Fig 1).

For each hypothetical uptake scenario, to calculate the potentially averted events, multiple season-specific input parameters (2016/2017, 2017/2018, 2018/2019) were collected from reports: the vaccine effectiveness, the size of the Dutch teacher population, the registered teacher self-reported influenza sick leave and duration, and the cumulative incidence of

---

**INFLUENZA-RELATED EVENTS**

1. **Estimated total notifications of influenza absenteeism (per year):**
   teacher population (26, 27) * sick leave frequency (28) * percentage self-reported flu as reason for sick leave (29-31) * rate of specimens testing positive for influenza from influenza-like illness GP visits (32-34)

2. **Estimated total number of days of influenza absenteeism (per year):**
   teacher population (26, 27) * sick leave frequency (28) * average number of days per sick leave (28) * percentage self-reported flu as reason for sick leave (29-31) * rate of specimens testing positive for influenza from influenza-like illness GP visits (32-34)

3. **Estimated total number of influenza GP visits:**
   teacher population (26, 27) /10,000 * incidence of GP visits for influenza-like illness in working age population per 10,000 inhabitants (21) * rate of specimens testing positive for influenza from sampled patients with influenza-like illness who visited a GP (32-34)

**POTENTIALLY AVERTED INFLUENZA-RELATED EVENTS**

For each hypothetical vaccination uptake scenario (2%, 10%, 25%,50% and 70% uptake) we calculated:

**Number of potentially averted events = N * vaccine effectiveness * vaccination uptake.**

**NUMBER NEEDED TO VACCINATE TO PREVENT ONE EVENT**

**1 / (vaccine effectiveness * N / population size)**

N = the estimated total number of events without the hypothetical vaccination programme as calculated in the above bullets 1 to 3. For Amsterdam (the only city with a teacher vaccination program) an adjusted formula was used to account for the actual uptake in the 2% scenario (see Fig 1 in supplement S5).

**Fig 1. Calculation of events, potentially averted events, and number needed to vaccinate to prevent one event [21, 26–34].**

---

medically attended influenza-like illness in adults in primary care [19–34]. Because influenza sick leave and influenza-like illness can be caused by respiratory pathogens other than influenza, we further adjusted these rates by multiplying them by the proportion of specimens testing positive for influenza from sampled patients with influenza-like illness who visited a GP [32–34]. This proportion was from the total Dutch population, as teacher-specific data was not available. Separate season-specific parameters were used because the severity of influenza epidemics varies from year to year [3, 4]. Additionally, teacher populations, sick leave, and vaccine effectiveness can vary from year to year. We distinguished between primary and secondary school scenarios. In the Netherlands primary education covers ages 4–11 years (including kindergarten for 2 years and the following 6 years of classes, all in the same school), and secondary education covers ages 12 years and up (with a duration of 4 to 6 years of classes, depending on the school level).

We calculated the number of events potentially averted by influenza vaccination in the total teacher population of the Netherlands and separately for the two large cities: Amsterdam and Rotterdam.

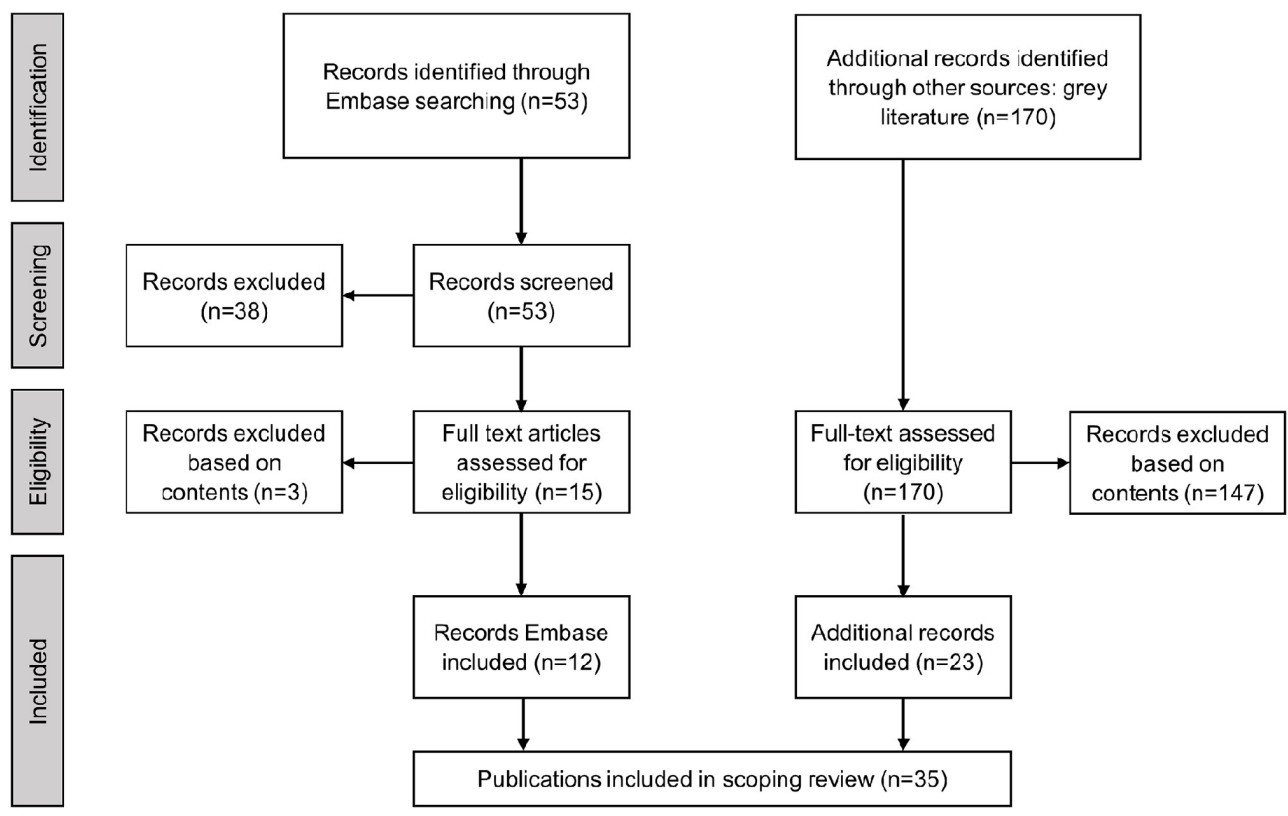

**Fig 2. Flow diagram of publication selection process.**

## Results

### Scoping review

In total, 12 scientific articles and 23 grey publications were included in the final scoping review (Fig 2). S1 and S2 Files contain detailed descriptions, tables and summaries of *Characteristics*, *Implementation details of vaccination*, *Vaccination uptake*, *Teachers' attitudes*, and *Impact*. In short, the 12 scientific articles were published between 2007 and 2019, mostly conducted in the USA, and were mostly questionnaire/survey-based. Only one scientific article reported the impact of influenza vaccination on teacher absenteeism [35]; the authors found that those who were vaccinated during a period of four months in 2007 had lower absenteeism than those who remained unvaccinated during that period (0.7% absenteeism vs 3.6%, p<0.001, on a study population size of 98, Gold Coast, Queensland, Australia), but they found no effect in 2009 when the vaccine viruses did not match the circulating viruses (see S1 File for additional results).

The 23 grey publications (published March 2007—February 2020) included little description of impact. News articles mentioned the offer of free influenza vaccination to reduce absenteeism [36], with one article stating that 75 is the number needed to vaccinate to prevent one case in healthy adults [37] (see additional results in S2 File).

Detailed results of interviews with 10 key informants are described in S3 File. Half of respondents expected the effectiveness of influenza vaccination for teachers to be low. The scientific evidence for the effect of vaccinating teachers was mentioned to be lacking but deemed crucial to be known, also by 50% of the respondents. The Dutch newspaper search resulted in 56 unique relevant articles (see S4 File for detailed results, Tables S4.2-S4.7 and Figures S4.1

and S4.2). The first newspaper article was published in February 2015, describing schools' problems arranging for substitute teachers. The next reports followed two years later in a Belgian newspaper, and from February 2018 onwards, the number of reports in Dutch newspapers increased up to 22, 21 and 10 in season-years 2017/2018, 2018/2019 and 2019/2020, respectively. The overall sentiment of the articles was mainly positive (n = 30, 53.6%, excluding duplicates) regarding vaccinating schoolteachers for influenza, but 26.8% had no clear opinion, and 17.9% were negative (unknown: 1.8%).

## Estimating influenza-related events

**Estimated influenza absenteeism.** In the three influenza seasons, teachers in primary education had a lower incidence of absenteeism notifications (range 1.0–1.2 per teacher per year) than those in secondary education (1.6–1.8 per teacher per year) [28] (Table 1). This data

**Table 1. Input parameters found for calculating the prevalence of teacher influenza events in the Netherlands nationally[*].**

| INPUT PARAMETERS | 2016/2017 | 2017/2018 | 2018/2019 | 2019/2020 |
|---|---|---|---|---|
| Teacher population PE[1] | 127,088 | 127,006 | 128,275 | 129,022 |
| Teacher population SE[2] | 75,585 | 75,973 | 75,694 | 75,284 |
| Absenteeism notifications PE (per teacher per year)[3] | 1.2 | 1.0 | 1.1 | n.a. |
| Absenteeism notifications SE (per teacher per year)[3] | 1.8 | 1.6 | 1.8 | n.a. |
| Average duration per absenteeism PE (days per absence)[3] | 3.4 | 3.3 | 3.4 | n.a. |
| Average duration per absenteeism SE (days per absence)[3] | 2.6 | 3.3 | 3.6 | n.a. |
| Self-reported flu as reason for last absence (%)[4, 5, 6] | 39.9 | 39.6 | 39.9 | n.a. |
| Influenza-like illness GP visits (age 15–44) (per year per 10,000)[**7] | 134 | 171 | 109 | n.a. |
| Influenza-like illness GP visits (age 45–64) (per year per 10,000)[**7] | 177 | 154 | 143 | n.a. |
| Average influenza-like illness GP visits (age 15–64) (per year per 10,000)[**] | 155.5 | 162.5 | 126 | n.a. |
| Rate of specimens testing positive for influenza from influenza-like illness GP visits (age 15–64)[7] | 0.41 | 0.64 | 0.42 | n.a. |
| I-MOVE+ pooled VE against H1N1 (age 15–64)[8] | n.a. | 50 (CI: 28 to 66) | 49 (CI: 29 to 64) | n.a. |
| I-MOVE+ pooled VE against H3N2 (age 15–64)[8] | 33.6 (CI: 17.9 to 46.3) | 33 (CI: -3 to 56) | -26 (CI: -66 to 4) | n.a. |
| I-MOVE+ pooled VE against B (age 15–64)[8] | n.a. | 21 (3 to 36) | n.a. | n.a. |
| Proportion H1N1/H3N2/B in Netherlands (all ages)[7,9] | 0.01/0.96/0.03 | 0.16/0.18/0.66 | 0.53/0.47/0.01 | 0.42/0.49/0.075 |
| Weighted overall VE against influenza confirmed ILI for NL[7,8,9] | 32.3 | 27.8 | 13.75 | n.a. |

n.a. = not available (for VE: not available for influenza subtypes that hardly circulated in that year).

VE = vaccine effectiveness

PE = primary education

SE = secondary education

GP = general practitioner

[*] = for Amsterdam and Rotterdam specific input parameters, see S5 File Tables S5.2 and S5.3

[**] = in respiratory season (week 40 –week 20, with the average being the simple average of the incidence in those aged 15–44 and 45–64 years)

[1] Report DUO: onderwijspersoneel PO in personen 2011–2019 [27];

[2] Report DUO: onderwijspersoneel VO in personen 2011–2019 [26];

[3] Report DUO: verzuimkengetallen 2016–2018 [28];

[4, 5, 6] Report CBS: nationale enquête arbeidsomstandigheden 2016, 2017 and 2018 [29–31];

[7] Report RIVM: annual surveillance of influenza and other respiratory infections in the Netherlands [21].

[8] I-MOVE end of season pooled influenza vaccine effectiveness report (2017 and 2018) [19, 20].

[9] NIC, Nivel Nieuwsbrief influenza-surveillance 2016–2017, 2017–2018, 2018–2019 and 2019–2020 [22–25].

**Table 2. Teacher population size and estimated† number of events per influenza-related event in teachers (absenteeism, physician visits).**

| | Netherlands | | Amsterdam | | Rotterdam | |
|---|---|---|---|---|---|---|
| | min‡ | max‡ | min‡ | max‡ | min‡ | max‡ |
| **Teacher population primary and secondary education**** | 202,673 | 203,969 | 9,357 | 9,428 | 7,627 | 7,710 |
| **Estimated notifications of influenza absenteeism (per year)*** | 46,479 | 62,996 | 2,520 | 3,542 | 1,713 | 2,356 |
| **Estimated total days of influenza absenteeism (per year)*** | 141,262 | 207,886 | 7,674 | 11,689 | 4,223 | 6,007 |
| **Estimated total influenza GP visits (per year)** | 1,079 | 2,111 | 50 | 97 | 40 | 78 |

* Teacher self-reported flu further adjusted for proportion of influenza positive specimens from adult influenza-like illness in patients in the national influenza surveillance.

** Total teacher population size available from 'Dienst Uitvoering Onderwijs' [26, 27].

† See S5 File for estimation details.

‡ Minimum and maximum value of the 3 estimations from the 3 years 2016/2017 to 2018/2019.

comprised overall absenteeism due to illness, but excluded absenteeism due to other reasons like maternity leave, study leave and emergency leave. Between 39.5% and 40.0% of the self-reported reasons for illness absenteeism was self-ascribed to influenza infection [29–31] (Table 1). The rate of positive test results for influenza virus obtained at GP visits for influenza-like illness was between 41% and 64%, depending on the season [21] (Table 1). Thus, the final calculation totaled 46,479–62,996 influenza absenteeism notifications per year in the total teacher population in the Netherlands (23–31% of total teacher population) (Table 2). Incidence of absenteeism notifications in Amsterdam was higher in both primary education (1.2–1.4 per teacher per year) and secondary education (1.9–2.2 per teacher per year) than it was nationally, while the incidence in Rotterdam was comparable with national numbers (1.0–1.2 per teacher per year in primary education and 1.6–1.7 per teacher per year in secondary education) (see S5 File, Tables S5.2 and S5.3 for all specific input parameters for Amsterdam and Rotterdam).

**Estimated influenza absenteeism duration.** Absenteeism duration is measured yearly in almost all Dutch teachers (coverage, 97–98% in primary education and 94–96% in secondary education 2016–2018). The average duration ranged from 20–22 days primary education and 12–14 days secondary education across the three seasons of 2016–2018 [28]. As this duration comprised overall sick leave, long-term sick leave skews this reported average upward. On the assumption that influenza infection causes only short-term sick leave, we adjusted these crude estimates by including only sick leave of ≤1 week's duration with use of the reported survival tables. This adjustment resulted in a lower average duration of 3.3–3.4 days for primary education and 2.6–3.6 days for secondary education (Table 1). When sick leave of ≤2 weeks' duration was included, the average was roughly 1.5 days higher. Separate survival tables were not reported for Amsterdam and Rotterdam. With their respective crude averages (Rotterdam: 17–18 days primary education and 9–10 days secondary education and Amsterdam: 18–22 days primary education, 13–14 days secondary education), we assumed the same underlying national duration distribution. We then calculated the two cities' respective average short-term absenteeism (Rotterdam: 2.6–3.0 days primary education and 1.9–2.6 days secondary education and Amsterdam: 2.7–3.3 days primary education and 2.8–3.6 days secondary education) (Tables S5.2 and S5.3, S5 File). Multiplication of the estimated incidence of absenteeism notifications by the average days per sick leave gave an estimated total of 141,262–207,886 days of influenza-related sick leave per year in the teacher population in the Netherlands (Table 2).

**Estimated total number of influenza-related GP visits.** The incidence of GP visits for influenza-like illness in the working-age population per 10,000 inhabitants is calculated yearly

[21]. During the study period, nationally, this incidence ranged between 109 and 171 per 10,000 inhabitants in groups aged 15–44 years and 45–64 years (groups roughly overlapped with work-force age; teacher-specific data were unavailable). We further adjusted this incidence for the rate of positive testing for influenza at GP visits [26, 27, 32–34]; this resulted in an annual estimated 1,079–2,111 influenza-related GP visits by teachers nationally, and an estimated 50–97 visits in Amsterdam, and 40–78 visits in Rotterdam (Tables S5.2 and S5.3, S5 File).

### Estimating potentially averted influenza-related events

To calculate numbers of influenza-related events potentially averted by vaccination, we searched input parameters on vaccine effectiveness and set several hypothetical scenarios of vaccination uptake.

1. *Vaccine effectiveness*: Vaccine effectiveness in Europe is available from the I-MOVE project, pooling data from 11 countries (Table 1). The I-MOVE+ report of 2016–17 reported a vaccine effectiveness for only A(H3N2), as it was the main circulating strain. The vaccine effectiveness was 33.6% against laboratory-confirmed influenza subtype in the group aged 15–64 years [19]. The I-MOVE+ report of 2017–18 reported vaccine effectiveness's for the group aged 15–64 of 50%, 33% and 21% against influenza subtypes A(H1N1)pdm09, A(H3N2) and influenza B, respectively [20]. For the 2018–19 season, interim I-MOVE results were used. A vaccine effectiveness of 49% and -26% against influenza subtypes A(H1N1) and A(H3N2), respectively, were reported in the report of surveillance of influenza in the Netherlands [21]. The National Influenza Centre (NIC) and the Netherlands Institute for Health Services Research (Nivel) reported proportions of influenza subtypes H1N1/H3N2/B in all ages of 0.01/0.96/0.03, 0.16/0.18/0.66, 0.53/0.47/0.01 and 0.42/0.49/0.075 for the seasons 2016/17, 2017/18, 2018/19 and 2019/20, respectively [22–25].

2. *Vaccination uptake scenarios*. Vaccination impact in hypothetical scenarios with a vaccination uptake of 2%, 10%, 25%, 50% and 70% was calculated with input of parameters reported for seasons 2016/2017-2018/2019. The two highest uptake scenarios were included, as this uptake was observed for one Dutch school board and reported in the USA [35, 38–47]. The lowest scenario was included, as this 2% uptake was observed for two years in Amsterdam [48, 49]. Importantly, the vaccination program was little known among many teachers (see interview results, S3 File).

**Estimated number of influenza-related events averted.** The estimated number of potentially averted events was calculated for the entire country, and separately for Amsterdam and Rotterdam (Table 3, and results per separate season given in S5 File, Tables S5.4-S5.6). In Amsterdam, a vaccination uptake of 2% in 2016/17 to 2018/19 would have averted 0–1 GP visits, 7–20 notifications of influenza absenteeism and 23–65 days of influenza absenteeism across those three seasons (Table 3). Had the vaccination uptake been 50%, it would have averted an estimated 3–14 GP visits, 185–492 notifications of influenza absenteeism and 585–1,625 days of influenza absenteeism across the three seasons (239–793 days in primary education; 346–832 days in secondary education). In Rotterdam, estimated potentially averted numbers would have been lower at 50% uptake, with 120–327 absenteeism notifications and 324–835 absent days, owing to the smaller teacher population and slightly lower sick leave numbers (Table 3). Nationally, at 2% vaccination uptake in 2018/2019, this estimate would have been 3–12 GP visits, 128–350 absenteeism notifications and 447–1,156 absent days; at 50% uptake, the estimate would have been 74–293 GP visits, 3,195–8,756 absenteeism notifications and 11,178–28,896

**Table 3. Estimated number of averted influenza events (NAE) in teachers by hypothetical vaccine uptake scenarios.**

|  | NETHERLANDS | | AMSTERDAM | | ROTTERDAM | |
|---|---|---|---|---|---|---|
| | **Estimated** NAE (notifications of influenza absenteeism) | | | | | |
| **Hypothetical vaccination uptake scenario** | min | max | min | max | min | max |
| **2%** | 304 | 735 | 18 | 40 | 11 | 27 |
| **10%** | 1522 | 3677 | 88 | 198 | 57 | 135 |
| **25%** | 3804 | 9191 | 220 | 496 | 143 | 337 |
| **50%** | 7608 | 18383 | 439 | 991 | 286 | 674 |
| **70%** | 10651 | 25736 | 615 | 1388 | 400 | 943 |
| | **Estimated** NAE (total days of influenza absenteeism) | | | | | |
| **Hypothetical vaccination uptake scenario** | min | max | min | max | min | max |
| **2%** | 5499 | 11933 | 278 | 636 | 164 | 360 |
| **10%** | 27496 | 59663 | 1390 | 3180 | 818 | 1802 |
| **25%** | 68739 | 149159 | 3476 | 7950 | 2044 | 4504 |
| **50%** | 137478 | 298317 | 6952 | 15900 | 4088 | 9009 |
| **70%** | 192470 | 417644 | 9733 | 22260 | 5723 | 12612 |
| | **Estimated** NAE (influenza GP visits) | | | | | |
| **Hypothetical vaccination uptake scenario** | min | max | min | max | min | max |
| **2%** | 3 | 12 | 0 | 1 | 0 | 0 |
| **10%** | 15 | 59 | 1 | 3 | 1 | 2 |
| **25%** | 37 | 147 | 2 | 7 | 1 | 5 |
| **50%** | 74 | 293 | 3 | 14 | 3 | 11 |
| **70%** | 104 | 411 | 5 | 19 | 4 | 15 |

Min/max: the lowest (minimum) and highest (maximum) estimate observed within the 3 calculated seasons (2016/2017-2018/2019).

absent days (Table 3). Total averted influenza-related absenteeism days did not differ greatly by primary and secondary education (5,527–14,765 averted days in primary education; 5,651–14,131 averted days in secondary education) because the smaller secondary-teacher population was offset by higher absenteeism rates. Independent of location and event, a vaccination uptake of 2% was estimated to result in 0.3–0.6% prevented events, calculated by dividing the estimated number of averted events by the total estimated number of events. For a vaccination uptake of 50%, this number was estimated to increase to 6.9–16.1% prevented events (Table 4). Our results translate to 11.6–31.9 vaccinations required to prevent one teacher sick leave

**Table 4. Prevented proportion\* of all influenza events in teachers by different vaccine uptake scenarios.**

| Hypothetical vaccination uptake scenario | Prevented proportion | |
|---|---|---|
| | min (%) | min (%) |
| **2%** | 0,3 | 0,6 |
| **10%** | 1,4 | 3,2 |
| **25%** | 3,4 | 8,1 |
| **50%** | 6,9 | 16,1 |
| **70%** | 9,6 | 22,6 |

\* Estimated number of averted events divided by total estimated number of events. These are independent of location and type of influenza event (absenteeism notifications, or duration, or GP visits).

Min/max: the lowest (minimum) and highest (maximum) estimate observed within the 3 calculated seasons (2016/2017-2018/2019).

**Table 5. Number needed to vaccinate to prevent one event.**

| | NETHERLANDS | | AMSTERDAM | | ROTTERDAM | |
|---|---|---|---|---|---|---|
| | min | max | min | max | min | max |
| Number needed to vaccinate to prevent one notification influenza absenteeism | 11.6 | 31.9 | 9.5 | 25.5 | 11.5 | 31.4 |
| Number needed to vaccinate to prevent one working day lost due to influenza absenteeism | 3.5 | 9.1 | 2.9 | 8.0 | 4.5 | 11.6 |
| Number needed to vaccinate to prevent one influenza GP visit | 346 | 1374 | 346 | 1374 | 346 | 1374 |

Min/max: the lowest (minimum) and highest (maximum) estimate observed within the 3 calculated seasons (2016/2017-2018/2019).

notification or 3.5–9.1 vaccinations to prevent one day of teacher absenteeism in the seasons 2016/2017 to 2018/2019 (Table 5).

## Discussion

Our study shows a paucity of scientific publications on teacher influenza vaccination, while public interest on the topic has increased. If vaccine uptake were to be moderate to high, impact on sick leave could be considerable.

Of 12 scientific papers, only a study in Australia reported the impact of vaccinating school teachers for influenza; vaccinated teachers had lower rates of absenteeism than those who were unvaccinated. The other 11 papers mainly assessed teachers attitudes and uptake, mostly in the USA. Grey literature showed that a few European countries nationally recommend influenza vaccination for teachers. In the Netherlands, no municipality, except Amsterdam in 2018 and 2019, was found to offer free influenza vaccination to teachers. However, multiple Dutch school boards do offer vaccination to their personnel, according to key informants and some newspaper articles. The main motivation for the offer was a reduction in absenteeism, but vaccine uptake levels were unreported. Unfortunately, details on how and where vaccinations were provided were either not known or not given. From February 2018, amidst a heavy influenza season, Dutch newspaper reporting on this topic increased; often reporting a positive attitude towards vaccination. However, the vaccination campaign among teachers in Amsterdam resulted in a vaccination uptake of only 2%, which was lower than expected. Reportedly, this low result could have occurred because the plans for vaccine administration were made hastily. Vaccinations were offered on only 2 evenings at one health centre, and the reach of the teacher population was limited. Interviewees expected that if communication and vaccination accessibility were to improve that vaccination uptake by teachers would increase. However it is unclear what the uptake would have been if Amsterdam had reached out to all teachers directly or had provided at-school vaccination.

Some key informants viewed financial incentives as unnecessary or undesirable, but these incentives were reported in grey literature to increase vaccination uptake. While vaccination uptake and its predictors have been studied in in a different sector (health care workers [50, 51]), Dutch teachers' knowledge of influenza vaccination and their attitudes regarding it are unknown. Future research could specifically address teachers through survey or interviews to understand their attitudes, concerns and practical barriers regarding influenza vaccination. Additionally, a pilot study offering free influenza vaccination on location to pre-informed teachers in several schools could provide actual teacher participation rates in an easily accessible setting and might provide key lessons for new or different vaccination campaigns. Many key informants expressed that they needed to know the potential impact of teacher influenza vaccination Future research should also focus on estimating the effect on sick leave of vaccinating teachers. Our impact calculations provide an initial estimate of influenza sick leave that

might be averted by vaccinating the teacher population. The reduction in sick leave could be considerable with a moderate to high vaccination uptake. Future research should additionally focus on the cost-effectiveness of teacher vaccination both for schools and for public health generally. Ideally it would also consider the potential effects of easier accessibility and financial incentives on the uptake.

In the Netherlands, local teacher shortages were the main motivation for offering free influenza vaccination to school teachers (in Amsterdam, and by few school boards elsewhere). The educational sector showed a higher-than-average absenteeism, with an increasing percentage due to influenza infection [29–31]. The largest teacher shortages were in the western part of the Netherlands [10, 11], the same region with the largest number of newspaper articles about teacher influenza vaccination. Teacher shortages have been increasing, and the largest increase shown in online vacancies was in 2017/2018, a year with a very severe influenza season [52]. The number of vacancies increased by 50% compared to the previous year for personnel in primary education and by almost 25% in all education sectors combined [10]. In 2019, teacher shortages were 3.4%, 2.7% and 2.0% in the large cities of Amsterdam, Rotterdam and The Hague, respectively. In 2020 these numbers were expected to reach 8.4%, 6.6% and 4.7%, respectively [11]. In these three cities, requests have come from either the council's education department (Amsterdam) or a political party (Rotterdam and The Hague) to set up a vaccination programme for school teachers or to investigate the feasibility of such a plan. Dutch health care workers [51, 52] are offered free influenza vaccination to protect their vulnerable patients. But decreased absenteeism and uninterrupted care are considered additional benefits [53]. This outcome could be similar in the education sector for which potential uptake and predictors are not yet known.

A Cochrane review by Demicheli 2014 found that in healthy adults (not specifically school teachers and staff), at least 71 people required vaccination to prevent one laboratory-confirmed case of influenza (CI: 64 to 80) [54]. However, many influenza infections are usually not laboratory-confirmed. Their number is more optimistic than our estimate of 346–1374 vaccinations to prevent one teacher visit to a GP in the Netherlands. These are specifically visits for influenza-related illness, which we assume to be a relatively similar event to the laboratory confirmed cases in Demichelli's review. Laboratory testing and physician consultation might occur relatively late in the disease process. At the earlier disease stage of taking sick leave for influenza, our results indicate that only 3.5–9.1 vaccinated teachers were needed to prevent 1 day of teacher influenza sick leave. These are relevant results, as there is uncertainty about the effect of vaccines on working days lost [5]. Our relatively high estimates of potentially averted absenteeism in teachers may be likely, given the conclusion of Nichol et al (1995). They reported that immunization decreased absenteeism from work due to upper respiratory illness by 43% and absenteeism due to all illnesses by 36% in healthy, working adults [55]. Saxen et al (1999) studied the effect of influenza vaccination in health care providers working in paediatric settings. They found a reduction of 28% in absenteeism related to respiratory infections [56]. Another review study of health care workers concluded that absenteeism was less frequent and shorter in those vaccinated, but the magnitude of this reduction was unavailable [57].

In school children, influenza vaccination showed a positive effect on overall influenza morbidity, not only in schools [58] but also in the community [43, 59]. For the 2009 H1N1 influenza pandemic, it was found that vaccinating school children decreased absenteeism of both teachers and students [60].

Whether teachers have a greater risk of influenza infection is not clear. Teachers possibly are at higher risk of *head/chest colds* compared to other workers [61], but these may not be caused solely by influenza viruses. In contrast, Elizondo-Montemayor, et al. showed the

prevalence of influenza A(H1N1pmd09) antibodies did not differ between school teachers (elementary and middle school) and the general population. The antibody prevalence was even lower in high school teachers than in the general population [62].

Calculating the potential impact of vaccination comes with some difficulties. One pitfall of our impact calculations is that we did not calculate additional effects. For instance, one report showed that 77% of the school employees with symptoms of influenza-like-illness did not report being sick and worked while ill [39]. Vaccination would also reduce this phenomenon, but we could not calculate this potential impact due to lack of Dutch data. Additionally, we did not account for the herd immunity that might occur with higher vaccination uptake, resulting in a larger impact than we estimated. For Amsterdam, we assumed that none of the teachers, other than those vaccinated at the public health service, had received influenza vaccination from their GP in that year within the national vaccination programme for risk groups. But, some teachers probably had received vaccination from their GP because of age (60+) or underlying medical conditions (within the general immunization programme). The actually registered sick leave that we used as input parameter would have been higher had they not been vaccinated within the general immunization programme. Therefore, we expect that we have underestimated, rather than overestimated, impact of vaccinating teachers. The vaccination campaign in Amsterdam seemed little known among teachers, and vaccination uptake was expected to be higher if all teachers had been informed. In health care workers, easier access to vaccination and education programmes on influenza vaccination were reported to increase vaccination uptake [63]. Both might also increase the vaccination uptake in teachers. Higher vaccination uptake would result in a higher estimated impact. To avoid overestimation of the number of days of sick leave due to influenza infection, we assumed influenza caused only short-term sick leave of ≤7 days. Thus we had included only short sick leave reported by teachers as an input parameter in our estimation. This might have resulted in an underestimation of impact, as influenza can also cause sick leave >7 days. However, only 4–5% of all sick leave was reported as lasting 8–14 days. We performed straightforward impact calculations. Future study could incorporate the scenarios of uncertainty in the various input data and thus improve the estimates by providing a range for the expected impact. The proportion of teachers' self-reported cases of influenza that were actually caused by the influenza virus needs investigation. For lack of data, we assumed it was the same as the proportion of visits to GP's for influenza-like illness that were confirmed as caused by the influenza virus (known for the general population, not for teachers specifically).

The current COVID-19 pandemic could lead to increased attention focused on seasonal influenza vaccination in teachers. The influenza H1N1 pandemic in 2009 also resulted in multiple countries adding risk groups to their national recommendations for the influenza vaccination [64] and increased seasonal influenza vaccine uptake in the 2009/2010 season [65]. Additionally, increasing seasonal influenza vaccine uptake to reduce the burden of the disease is a key focus of the EU and World Health Organization in preparation for COVID-19 resurgences [66]. Also, the management of pressure in the education sector that has been burdened already with staff shortages may receive increasing attention.

In conclusion, international scientific literature on influenza vaccination of teachers remains sparse and vaccination of this group is not nationally advised in most countries. However, Dutch media attention is growing, sparked by teacher shortages in recent years. The impact of vaccinating teachers has been particularly understudied. This is the case despite some key experts and media have reported that any school board or other entity that is considering offering influenza vaccination to teachers needs to understand the impact of the vaccination. Our estimates showed that vaccinating teachers against influenza might be associated with a substantial decrease of sick leave days due to that viral infection. Teacher surveys, in-

depth cost-effectiveness studies and a pilot at-school influenza vaccination programme could provide critical information about teacher vaccination.

## Supporting information

**S1 File. Scientific literature review.**
(DOCX)

**S2 File. Grey literature review.**
(DOCX)

**S3 File. Interviews.**
(DOCX)

**S4 File. Newspaper monitoring.**
(DOCX)

**S5 File. Impact calculations.**
(DOCX)

**S6 File. Prisma checklist scoping review.**
(DOCX)

## Acknowledgments

We thank all key experts who took part in the interviews.

## Author Contributions

**Conceptualization:** Brigitte van Cleef, Aimée Tjon-A-Tsien, Wim van der Hoek, Liselotte van Asten.

**Data curation:** Anne Huiberts.

**Formal analysis:** Anne Huiberts.

**Investigation:** Anne Huiberts.

**Methodology:** Anne Huiberts, Brigitte van Cleef, Aimée Tjon-A-Tsien, Frederika Dijkstra, Liselotte van Asten.

**Supervision:** Brigitte van Cleef, Aimée Tjon-A-Tsien, Liselotte van Asten.

**Visualization:** Anne Huiberts.

**Writing – original draft:** Anne Huiberts.

**Writing – review & editing:** Anne Huiberts, Brigitte van Cleef, Aimée Tjon-A-Tsien, Frederika Dijkstra, Imke Schreuder, Ewout Fanoy, Arianne van Gageldonk, Wim van der Hoek, Liselotte van Asten.

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
