## [Decision Letter · Decision Letter 0]

15 Mar 2021

PONE-D-20-40234

Influenza vaccination of school teachers: a scoping review and an impact estimation

PLOS ONE

Dear Dr. Huiberts,

Thank you for submitting your manuscript to PLOS ONE. After careful consideration, we feel that it has merit but does not fully meet PLOS ONE’s publication criteria as it currently stands. Therefore, we invite you to submit a revised version of the manuscript that addresses the points raised during the review process.

We look forward to receiving your revised manuscript.

Kind regards,

Shinya Tsuzuki, MD, MSc

Academic Editor

PLOS ONE

Journal Requirements:

3. Our staff editors have determined that your manuscript is likely within the scope of our Call for Papers on Influenza. This editorial initiative is headed by PLOS ONE Guest Editors Dr. Meagan Deming and Dr. Deshayne Fell. The Collection encompasses research on influenza prevention on every level, including in vitro, translational, behavioral, and clinical studies; disease and immunity modelling; as well as new approaches to influenza prevention. Additional information can be found on our announcement page: https://collections.plos.org/call-for-papers/influenza/.

Currently, your manuscript is included in the group of papers being considered for this call. Please note that being considered for the Collection does not require additional peer review beyond the journal’s standard process and will not delay the publication of your manuscript if it is accepted by PLOS ONE. We would greatly appreciate your confirmation that you would like your manuscript to be considered for this Collection by indicating this in your next cover letter. If you would prefer to remove your manuscript from collection consideration, please specify this in your cover letter.

4. Please provide additional details regarding participant consent. In the ethics statement in the Methods and online submission information, please ensure that you have specified whether consent was informed.

5. We note that Figure 3 in your manuscript and Figure 2 in S4 file contain map images which may be copyrighted. All PLOS content is published under the Creative Commons Attribution License (CC BY 4.0), which means that the manuscript, images, and Supporting Information files will be freely available online, and any third party is permitted to access, download, copy, distribute, and use these materials in any way, even commercially, with proper attribution. For these reasons, we cannot publish previously copyrighted maps or satellite images created using proprietary data, such as Google software (Google Maps, Street View, and Earth). For more information, see our copyright guidelines: http://journals.plos.org/plosone/s/licenses-and-copyright.

(1) You may seek permission from the original copyright holder of Figure 3 and Figure 2 in S4 file to publish the content specifically under the CC BY 4.0 license. 

Additional Editor Comments:

In this time, the decisions made by reviewers were split, but I believe that the points both reviewers mentioned can be addressed by appropriate revision. At present the manuscript includes too many kind of methodologies, and as a result, it becomes distracted to some extent. The scope of the manuscript should be clearer through its revision process.

Reviewers' comments:

Reviewer's Responses to Questions

**Comments to the Author**

1. Is the manuscript technically sound, and do the data support the conclusions?

Reviewer #1: No

Reviewer #2: Yes

2. Has the statistical analysis been performed appropriately and rigorously? 

Reviewer #1: Yes

Reviewer #2: Yes

3. Have the authors made all data underlying the findings in their manuscript fully available?

Reviewer #1: Yes

Reviewer #2: Yes

4. Is the manuscript presented in an intelligible fashion and written in standard English?

Reviewer #1: No

Reviewer #2: Yes

5. Review Comments to the Author

Reviewer #1: This manuscript presents the results of a scoping review and impact assessment of influenza vaccination of teachers in the Netherlands. Given the current discussion around COVID-19 vaccination of teachers and safely opening schools it is a highly relevant research question. Overall the article is clearly written and obviously represents a massive amount of work. However, I think there is too much information in this manuscript for one scientific article. I would recommend that the authors focus on the impact assessment, as the results of the scoping review (namely, there hasn't been much done in this field) are not particularly surprising. I suggest moving the scoping review methods and much of the results to a supplement (or splitting the manuscript into separate articles). I have attempted to provide some detailed comments by manuscript section, but these comments would likely be much more helpful to the authors once there has been a decision on how to proceed in regards to splitting of changing the focus of the current manuscript.

Introduction

Page 7 Line 102 - It is not universally true that there is no protection from year to the next. In fact, early life influenza infection has been shown to influenza susceptibility to later life infection This is particularly for H1N1pdm09 which has seen relatively little antigenic drift since it was established as a seasonal virus. Similarly, waning antibody level is another reason for annual vaccination campaigns.

Be specific about the protection afforded by seasonal influenza vaccines. The 30-50% number reported by CDC is against medically attended, symptomatic infection.

Page 7 line 117 - this phrasing is a little awkward. I would suggest saying that vaccination is associated with a reduction in absenteeism due to influenza associated illnesses.

Methods

The search strategy was clearly described. But why was January 2000 selected as a start date? Is this tied to Dutch vaccine recommendations or other criteria? If there is no rationale for the time restriction I would open it up to include any dates (with the expectation that very few studies were done prior to 2000). Of note, none of the included articles were published before 2007, this tracks with US vaccine recommendations which were expanded in the 2000s and early 2010s.

Results

Line 220 - Include reasons why scientific articles and grey publications were excluded - are the detailed descriptions of the 12 or the 15

Line 367 - You say absenteeism was reduced by too little to justify continuing the program according to a newspaper article. Was the actual reduction noted?

How was tone of the article (e.g. positive, indifferent) determined?

Line 387 - "we adjusted this number" - How was this adjustment done? What assumptions were made?

Discussion

Much of the first paragraphs in the discussion are repetition of what was presented in the scoping review section of the results.

Estimates of cases averted and absenteeism have been in the US, but don't seem to be mentioned in the discussion.

Overall

In my opinion, the scientific value of this paper is really in the impact assessment. I would reframe the narrative throughout to focus on that, minimizing the scoping review. This would mean a very brief summary of the scoping review methods and results, with most of the detail moved to the supplementary materials. In contrast I think much more detail is needed in the methods section for the impact assessment. Similarly the results section should focus primarily on the impact assessment instead of burying that material at the end after a detailed descriptions of the review.

Reviewer #2: Review

In this study, the authors conducted a comprehensive review on flu vaccination administered to school teachers and evaluated its requirements and rationale. I consider that this study’s objectives were (at least partially) achieved by using a combination approach of scoping review, interview, and impact estimation. However, this study is very complex because many issues and study methods are handled in the same paper. I believe this paper is worth publishing. However, some attempts to improve the readability are needed.

1. In my opinion, scoping review, interview (qualitative study), newspaper search, and impact estimation (modeling) seem different. Since the study consisted of synthesizing these four different studies, this manuscript looks like a policy document rather than an academic write-up. To clarify the scope of each study part, I suggest that the authors explain the aims and reasons for which this method was selected in every part.

2. In addition, the deductive theory flow in this study is as follows: 1. There is a lack of school teachers in Netherlands –> 2. The flu outbreaks worsen this situation –> 3. vaccine could save this situation –> 4. Currently, vaccination is not satisfactorily performed -> 5. So how can we do it? I consider this flow to be inconsistent in this manuscript in each part, leading to a decrease in readability. Therefore, I believe that being conscious of this flow can help readability.

3. Did the author know the low vaccine prevalence in Amsterdam before this survey was performed? If the authors knew this fact before the survey, I think they should mention the low vaccine prevalence in the Introduction section. This is because the low vaccine prevalence could explain the reason for this study and its rationale naturally.

4. In the Introduction section, the authors mentioned “overviewing any aspect vaccinating teachers against influenza,” however, the aim of the scoping review is identifying “gaps” between existing evidence and required evidence to state the specific purpose of scoping. Therefore, the authors should first define the aim of this scoping review. In this study, I understand that this scoping review aimed to collect evidence on the spread of flu vaccinations among school teachers.

5. Regarding this scoping review, did the authors identify regions with inadequate vaccination evidence based on the author’s purpose? This seems to be mentioned in the Conclusion section, although I could not find it in the main text.

6. The review of newspapers did not have an impact on the integrity of this study. Additionally, I do not know the scientific validity of these methods. Therefore, it may be better to remove this part from this study.

7. In the impact estimation part, statements that should be written in the Methods section are frequently seen in the Results section (model selection, way of adjustment, etc.). Please consider ameliorating this structure.

8. The conclusion is unclear. It could be understandable if the objectives of each part of this study are made intelligible.

6. PLOS authors have the option to publish the peer review history of their article (what does this mean?). If published, this will include your full peer review and any attached files.

Reviewer #1: No

Reviewer #2: **Yes: **Yoshiki Kusama

---

## [Author Response · Author response to Decision Letter 0]

23 Mar 2022

We thank the reviewers for their comments and suggestions. We believe they have greatly contributed to improving the quality and clarity of the work. Below, we provide a point by point response to all their points and to the additional PLOS ONE requirements.

PloS One additional requirements.

Reply: Thank you, we have checked the file names and all other PLOS ONE requirements.

REPLY: Checking the provided PLOS ONE link we have now changed this. We have made the data available in the manuscript, the anonimized excerpts of the interviews are in supplement S3 (pages 10-18).

REPLY: We checked our information letter and our consent form and we have now made the data available in the manuscript, see also point b below. 

REPLY: We have added the anonymized excerpts of the interviews in supplementary S3 (pages 10-18).

3. Our staff editors have determined that your manuscript is likely within the scope of our Call for Papers on Influenza. This editorial initiative is headed by PLOS ONE Guest Editors Dr. Meagan Deming and Dr. Deshayne Fell. The Collection encompasses research on influenza prevention on every level, including in vitro, translational, behavioral, and clinical studies; disease and immunity modelling; as well as new approaches to influenza prevention. Additional information can be found on our announcement page: https://collections.plos.org/call-for-papers/influenza/.

Currently, your manuscript is included in the group of papers being considered for this call. Please note that being considered for the Collection does not require additional peer review beyond the journal’s standard process and will not delay the publication of your manuscript if it is accepted by PLOS ONE. We would greatly appreciate your confirmation that you would like your manuscript to be considered for this Collection by indicating this in your next cover letter. If you would prefer to remove your manuscript from collection consideration, please specify this in your cover letter.

REPLY: we are certainly happy for publication within, or outside of this call, as PLOS ONE would wish. 

4. Please provide additional details regarding participant consent. In the ethics statement in the Methods and online submission information, please ensure that you have specified whether consent was informed. 

REPLY: We have added additional detail in the online submission form as well as to the methods section. For this we added the following sentences to supplement S3: 

‘Potential participants were approached by email and they received a letter asking whether they would agree to participate. The letter explained the research aims, interview procedures, and privacy matters (de-identification of transcripts). Written and oral consent was obtained from each consenting interviewee. The consent form included queries on whether the interviewee had read the information letter and whether all their queries and concerns had been addressed satisfactorily. Ethical approval was not required for this study according to Dutch legislation, as it concerned a once only interview.’ (Supplement S3, page 1-2, lines 26-38)

5. We note that Figure 3 in your manuscript and Figure 2 in S4 file contain map images which may be copyrighted. All PLOS content is published under the Creative Commons Attribution License (CC BY 4.0), which means that the manuscript, images, and Supporting Information files will be freely available online, and any third party is permitted to access, download, copy, distribute, and use these materials in any way, even commercially, with proper attribution. For these reasons, we cannot publish previously copyrighted maps or satellite images created using proprietary data, such as Google software (Google Maps, Street View, and Earth). For more information, see our copyright guidelines: http://journals.plos.org/plosone/s/licenses-and-copyright.

REPLY: We have now produced these map images using the free R statistical package, which is without copyright [R Core Team (2021). R: A language and environment for statistical computing. R Foundation for Statistical Computing, Vienna, Austria. https://www.R-project.org/]. The source data on province areas as available from Statistics Netherlands (CBS) and the Dutch land registry (Kadaster). Metadata are freely downloadable without copyright from the Dutch National Geo Registry: https://www.nationaalgeoregister.nl/geonetwork/srv/dut/catalog.search#/metadata/effe1ab0-073d-437c-af13-df5c5e07d6cd?tab=general).

To clarify these two sources for readers we have added the following subscript below the figure: ‘* province: data on province areas as available from Statistics Netherlands (CBS) and the Dutch land registry (Kadaster). Metadata are freely downloadable without copyright from the Dutch National Geo Registry: https://www.nationaalgeoregister.nl/geonetwork/srv/dut/catalog.search#/metadata/effe1ab0-073d-437c-af13-df5c5e07d6cd?tab=general). The figure is made using statistical software R [R Core Team (2021). R: A language and environment for statistical computing. R Foundation for Statistical Computing, Vienna, Austria. https://www.R-project.org/]’. 

Due to the reviewers other request on page 4 to focus more on the impact estimates and thus move much other text to the supplements, the map image of figure three has been removed from the main manuscript, and can be found in the supplement S4 (as figure S4.2). 

Additional Editor Comments:

In this time, the decisions made by reviewers were split, but I believe that the points both reviewers mentioned can be addressed by appropriate revision. At present the manuscript includes too many kind of methodologies, and as a result, it becomes distracted to some extent. The scope of the manuscript should be clearer through its revision process.

REPLY: We thank PLOS ONE for the thorough review. As both reviewers mention that the manuscript includes many methodologies, we followed the advice of both reviewers to move most parts of the scoping review to the supplements and to focus the main document more on the estimated impact sections. As suggested by the reviewers, a very short summary of methods and the main findings of the scoping review now remain in the main text, with reference to all scoping review details in supplements S1-S4. Then, in the manuscript we further elaborated on the methods and results of the impact estimates. This indeed clarifies the scope of the manuscript which is summarized in the abstract as follows: ‘This study describes current knowledge of influenza vaccination in teachers and estimates its potential impact’.

Reviewers' comments:

Reviewer's Responses to Questions

Comments to the Author

1. Is the manuscript technically sound, and do the data support the conclusions?

Reviewer #1: No

Reviewer #2: Yes

REPLY: To better clarify the scientific research we have expanded the description of the methods and results section of the impact estimation (as was also suggested by both reviewers in their separate comments) (page 9-10, lines 144-174 and page 11-19, lines 208-348). We have now also included an additional figure explaining the calculations (figure 2). 

 2. Has the statistical analysis been performed appropriately and rigorously? 

 Reviewer #1: Yes

Reviewer #2: Yes

REPLY: Thank you.

 3. Have the authors made all data underlying the findings in their manuscript fully available?

Reviewer #1: Yes

Reviewer #2: Yes

REPLY: Thank you.

 4. Is the manuscript presented in an intelligible fashion and written in standard English?

 Reviewer #1: No

Reviewer #2: Yes

REPLY: The manuscript has now been edited by a professional English language editor (S. Ebeling, freelance editor who has previously worked for The New England Journal of Medicine)

 5. Review Comments to the Author

Reviewer #1: This manuscript presents the results of a scoping review and impact assessment of influenza vaccination of teachers in the Netherlands. Given the current discussion around COVID-19 vaccination of teachers and safely opening schools it is a highly relevant research question. Overall the article is clearly written and obviously represents a massive amount of work. However, I think there is too much information in this manuscript for one scientific article. I would recommend that the authors focus on the impact assessment, as the results of the scoping review (namely, there hasn't been much done in this field) are not particularly surprising. I suggest moving the scoping review methods and much of the results to a supplement (or splitting the manuscript into separate articles). I have attempted to provide some detailed comments by manuscript section, but these comments would likely be much more helpful to the authors once there has been a decision on how to proceed in regards to splitting or changing the focus of the current manuscript.

REPLY: We appreciate the reviewer’s compliments regarding the relevance of the research question and the amount of work involved. We followed his/her advice to focus more on the impact assessment and to move much of the scoping review methods and results to the supplements (see supplements S1-S4). This is also in line with the same suggestion done by reviewer 2. A brief summary of the scoping review methods and results remain in the main manuscript. And we now elaborated much more on the methods and results section of the impact estimation. We also included an additional figure explaining the impact calculations (figure 2).

Introduction

Page 7 Line 102 - It is not universally true that there is no protection from year to the next. In fact, early life influenza infection has been shown to influenza susceptibility to later life infection This is particularly for H1N1pdm09 which has seen relatively little antigenic drift since it was established as a seasonal virus. Similarly, waning antibody level is another reason for annual vaccination campaigns. 

REPLY: We agree with the reviewer that this is not universally true. We have removed the section about a lack of year to year protection and we changed the sentence to: 

‘With influenza viruses constantly evolving (antigenic drift) [1] and waning antibody levels, a new vaccination campaign is needed annually [5].’ (page 6 lines 86-87)

Be specific about the protection afforded by seasonal influenza vaccines. The 30-50% number reported by CDC is against medically attended, symptomatic infection. 

REPLY: Thank you for noticing this omission, we have added that this is indeed against medically attended infection:

‘The seasonal vaccine offers partial (roughly 30-50%) protection against medically attended, symptomatic infection if the match between the vaccine viruses and the circulating viruses is strong [6]; it provides less protection if circulating strains deviate from predicted.’ (page 6 lines 87-90)

Page 7 line 117 - this phrasing is a little awkward. I would suggest saying that vaccination is associated with a reduction in absenteeism due to influenza associated illnesses. 

REPLY: We agree that the phrasing was awkward. We have now rephrased to: 

‘Influenza vaccination may also protect employees in sectors other than health care and might reduce absenteeism due to influenza-associated illnesses [9].’ (page 6 lines 103-104)

Methods

The search strategy was clearly described. But why was January 2000 selected as a start date? Is this tied to Dutch vaccine recommendations or other criteria? If there is no rationale for the time restriction I would open it up to include any dates (with the expectation that very few studies were done prior to 2000). Of note, none of the included articles were published before 2007, this tracks with US vaccine recommendations which were expanded in the 2000s and early 2010s.

REPLY: The selected period (2000-2020) was not tied to any policy or recommendations. As suggested by the reviewer we have opened up the time restriction to include any dates prior to 2000. This provided two additional scientific papers which after screening were not relevant (they were not about influenza vaccination in teachers). We adjusted the selected search period in the methods section (Page 8, S1, S2) and we adjusted figure 2 to reflect this search strategy change.

Results

Line 220 - Include reasons why scientific articles and grey publications were excluded - are the detailed descriptions of the 12 or the 15

REPLY: We agree with the reviewer that we had omitted this information. We have added the following text to the methods of the scientific and grey literature search:

‘Publications were excluded if they were not about influenza vaccination or not about teachers’. (supplement S1, page 1 lines 20-21, and supplement S2, page 1 lines 25-26) 

Line 367 - You say absenteeism was reduced by too little to justify continuing the program according to a newspaper article. Was the actual reduction noted?

REPLY: Unfortunately the newspaper item did not provide the actual reduction, which we now added to the sentence as follows: ‘The latter stopped offering teacher vaccination as absenteeism reduced too little. However, details on the magnitude of reduction and how and where vaccination was offered by these boards are not given.’ (supplement S4 page 2 lines 77-79)

How was tone of the article (e.g. positive, indifferent) determined?

REPLY: We have extended the methods with the following details about how the newspaper sentiment was determined: 

‘Positive sentiment was noted when a positive opinion was expressed, or when the article described actual implementation of vaccination of school teachers. A negative opinion or non-implementation were considered as a negative sentiment. When articles did not have a clear overall sentiment because a combination of positive and negative opinions was expressed in the article, we considered the sentiment “multiple”.’ (supplement S4, page 1, lines 32-36)

Line 387 - "we adjusted this number" - How was this adjustment done? What assumptions were made?

REPLY: This information was indeed missing. We have now added the explanation to the methods section as follows: 

‘Because influenza sick leave and influenza-like illness can be caused by respiratory pathogens other than influenza, we further adjusted these rates by multiplying them by the proportion of specimens testing positive for influenza from sampled patients with influenza-like illness who visited a GP [32-34]. This proportion was from the total Dutch population, as teacher-specific data was not available.’ (Page 9, lines 158-162)

Discussion

Much of the first paragraphs in the discussion are repetition of what was presented in the scoping review section of the results. 

REPLY: We certainly agree with the reviewer. The scoping review results have now been shortened to a very brief summary with all further details placed in supplements (S1-S4), as was suggested by both reviewers. This removes any repetition that was present in the body text. The main points now only remain discussed and placed into perspective in the discussion section. 

Estimates of cases averted and absenteeism have been in the US, but don't seem to be mentioned in the discussion. 

REPLY: These estimates are indeed available for the USA, albeit not specifically for teacher populations, and should indeed be mentioned. We have now added a reference to estimates in USA health care workers by referring to a review paper that reviewed multiple papers on this topic:

‘Also, a review study of health care workers concluded that the incidence of absenteeism due to influenza-like illness was reduced and its duration shortened in those vaccinated, but an estimate of the magnitude of reduction was unavailable [58].’ (Page 22, lines 422-425)

Overall

In my opinion, the scientific value of this paper is really in the impact assessment. I would reframe the narrative throughout to focus on that, minimizing the scoping review. This would mean a very brief summary of the scoping review methods and results, with most of the detail moved to the supplementary materials. In contrast I think much more detail is needed in the methods section for the impact assessment. Similarly the results section should focus primarily on the impact assessment instead of burying that material at the end after a detailed descriptions of the review.

REPLY: We agree with reviewer 2, whose comments are in line with reviewer 1. We adjusted the manuscript to focus mostly on the impact assessment and we moved much of the scoping review methods and results to the supplements (see supplements S1-S4). As the reviewer suggests, only a brief summary of the scoping review methods and results now remains in the main manuscript. In the manuscript we now elaborated much more on the methods and results of the impact estimation (page 9-10, lines 144-174 and page 11-19, lines 208-348). We also included an additional figure explaining the impact calculations (figure 2).

Reviewer #2: Review

In this study, the authors conducted a comprehensive review on flu vaccination administered to school teachers and evaluated its requirements and rationale. I consider that this study’s objectives were (at least partially) achieved by using a combination approach of scoping review, interview, and impact estimation. However, this study is very complex because many issues and study methods are handled in the same paper. I believe this paper is worth publishing. However, some attempts to improve the readability are needed.

REPLY: We thank the reviewer for their opinion that the paper is worth publishing if readability is improved. We made large adjustments to improve the readability. The scoping review (including the interviews) is now only briefly summarized in the body text (with reference to supplements S1-S4 for details), which greatly reduces the complexity of the manuscript. Additionally we elaborated the methods and results sections of the impact estimation to better clarify that topic (page 9-10, lines 144-174 and page 11-19, lines 208-348).

1. In my opinion, scoping review, interview (qualitative study), newspaper search, and impact estimation (modeling) seem different. Since the study consisted of synthesizing these four different studies, this manuscript looks like a policy document rather than an academic write-up. To clarify the scope of each study part, I suggest that the authors explain the aims and reasons for which this method was selected in every part.

2. In addition, the deductive theory flow in this study is as follows: 1. There is a lack of school teachers in Netherlands –> 2. The flu outbreaks worsen this situation –> 3. vaccine could save this situation –> 4. Currently, vaccination is not satisfactorily performed -> 5. So how can we do it? I consider this flow to be inconsistent in this manuscript in each part, leading to a decrease in readability. Therefore, I believe that being conscious of this flow can help readability.

REPLY: We certainly agree that the flow was not clearly explained in the manuscript. We therefore substituted the final paragraph of the introduction by a concise description of the flow:

‘Although Dutch media have recently reported on influenza vaccination of teachers as one way to decrease schoolteacher shortages, a comprehensive overview of literature is lacking. Therefore, we describe the current knowledge of teacher influenza vaccination using very broad input (scientific literature, grey literature, Dutch newspaper reports and information from key informants). Since knowledge of the impact of this vaccination in teacher populations is crucial for decision-making, but is lacking, we then estimated its potential impact.’ (Page 7, lines 116-121). We did not refer to reviewers points 4 and 5 as they were not part of the opinion or the scope of this manuscript. To better clarify this we added a sentence to the introduction:

‘Traditional risk groups targeted for influenza vaccination in many countries include the elderly (60+) and persons with underlying chronic conditions, because of their higher risk of influenza complications. In the Netherlands, this does not include teachers unless they are individually targeted because of fitting one of those traditional risk groups.’ (Page 6, lines 91-94). Also, touching on that topic, we do make a recommendation in the conclusion: ‘Teacher surveys, in-depth cost-effectiveness studies and a pilot at-school influenza vaccination programme could provide critical information about teacher vaccination.’ (Page 25, lines 485-486). 

3. Did the author know the low vaccine prevalence in Amsterdam before this survey was performed? If the authors knew this fact before the survey, I think they should mention the low vaccine prevalence in the Introduction section. This is because the low vaccine prevalence could explain the reason for this study and its rationale naturally. 

REPLY: Teachers are not a target group for influenza vaccination in the Netherlands. We agree that this was not very clearly stated and we have added the following to the introduction:

‘Traditional risk groups targeted for influenza vaccination in many countries include the elderly (60+) and persons with underlying chronic conditions, because of their higher risk of influenza complications. In the Netherlands, this does not include teachers unless they are individually targeted because of fitting one of those traditional risk groups.’ (Page 6, lines 91-94). 

An important reason for the study was the media attention for the potential of decreasing teacher shortages by influenza vaccination and one public health physician wondering if teacher vaccination would actually have an impact on teacher absenteeism. As little seemed known about influenza vaccination specifically in the teacher group we decided to perform a scoping review to provide an overview of the current knowledge on this topic and to identify knowledge gaps more accurately than from hearsay. The main knowledge gap that came forward was the lack of knowledge on the potential impact of teacher vaccination. Therefore we complemented the study with an impact estimation. We have now summarized this in the final paragraph of the introduction as follows: 

‘Although Dutch media have recently reported on influenza vaccination of teachers as one way to decrease schoolteacher shortages, a comprehensive overview of literature is lacking. Therefore, we describe the current knowledge of teacher influenza vaccination using very broad input (scientific literature, grey literature, Dutch newspaper reports and information from key informants). Since knowledge of the impact of this vaccination in teacher populations is crucial for decision-making, but is lacking, we then estimated its potential impact.’ (Page 7, lines 116-121).

4. In the Introduction section, the authors mentioned “overviewing any aspect vaccinating teachers against influenza,” however, the aim of the scoping review is identifying “gaps” between existing evidence and required evidence to state the specific purpose of scoping. Therefore, the authors should first define the aim of this scoping review. In this study, I understand that this scoping review aimed to collect evidence on the spread of flu vaccinations among school teachers.

REPLY: We agree that this final paragraph of the introduction was not clear. In the beginning of the introduction we added the information that in the Netherlands, teachers are not targeted for influenza vaccination. And we have replaced the unclear final paragraph by the following: ‘Although Dutch media have recently reported on influenza vaccination of teachers as one way to decrease schoolteacher shortages, a comprehensive overview of literature is lacking. Therefore, we describe the current knowledge of teacher influenza vaccination using very broad input (scientific literature, grey literature, Dutch newspaper reports and information from key informants). Since knowledge of the impact of this vaccination in teacher populations is crucial for decision-making, but is lacking, we then estimated its potential impact.’ (Page 7, lines 116-121). 

5. Regarding this scoping review, did the authors identify regions with inadequate vaccination evidence based on the author’s purpose? This seems to be mentioned in the Conclusion section, although I could not find it in the main text.

REPLY: Influenza vaccination for school teachers in not nationally recommended in the Netherlands. We agree that this is not entirely clear from the conclusion section. To clarify this, we have added a sentence in the introduction and we have now emphasized this in the conclusion section as follows (see bold/italic typing):

In the introduction:

 ‘Traditional risk groups targeted for influenza vaccination in many countries include the elderly (60+) and persons with underlying chronic conditions, because of their higher risk of influenza complications. In the Netherlands, this does not include teachers unless they are individually targeted because of fitting one of those traditional risk groups.’ (Page 6, lines 91-94).

In the conclusion:

However, most of the research was conducted in the USA, where influenza vaccination is already officially recommended for all citizens (excluding infants <6 months) and vaccination uptake is generally higher than in Europe. Grey literature showed that a few European countries nationally recommend influenza vaccination for teachers. In the Netherlands, no municipality, except Amsterdam in 2018 and 2019, was found to offer free influenza vaccination to teachers. However, multiple Dutch school boards do offer vaccination to their personnel, according to key informants and some newspaper articles. The main motivation for the offer was a reduction in absenteeism, but vaccine uptake levels were unreported. Unfortunately, details on how and where vaccinations were provided were either not known or not given. (page 20, lines 357-365)

6. The review of newspapers did not have an impact on the integrity of this study. Additionally, I do not know the scientific validity of these methods. Therefore, it may be better to remove this part from this study.

REPLY: The grey literature did not provide a full overview of the Dutch media attention for teacher influenza vaccination. Therefore we added a search of a Dutch newspaper database. We have removed almost all details on this topic from the manuscript to supplementary S4. This does indeed not impact the integrity of the main manuscript. As the media attention for influenza vaccination as a way to reduce teacher shortages was one of the reasons for this study we feel it best to still provide these details, but in a supplementary. 

7. In the impact estimation part, statements that should be written in the Methods section are frequently seen in the Results section (model selection, way of adjustment, etc.). Please consider ameliorating this structure. 

REPLY: We agree with the reviewer and we therefore moved those sentences to the methods section.

8. The conclusion is unclear. It could be understandable if the objectives of each part of this study are made intelligible. 

REPLY: We agree that this needed more clarity. To this end several sections were modified: the final paragraph of the introduction (to clarify the objectives), the first paragraph of the methods was shortened and simplified (also to clarify the objectives), and a few small clarifications were added to the final paragraph of the discussion:

In introduction:

‘Although Dutch media have recently reported on influenza vaccination of teachers as one way to decrease schoolteacher shortages, a comprehensive overview of literature is lacking. Therefore, we describe the current knowledge of teacher influenza vaccination using very broad input (scientific literature, grey literature, Dutch newspaper reports and information from key informants). Since knowledge of the impact of this vaccination in teacher populations is crucial for decision-making, but is lacking, we then estimated its potential impact.’ (Page 7, lines 116-121). 

In methods:

‘We conducted a scoping review of the considerations for and impact of influenza vaccination of school teachers. We then estimated the potential impact of teacher vaccination in the Netherlands at different scenarios of vaccine uptake for three influenza seasons (2016-2019).’ (Page 8, lines 123-125)’

In Discussion:

‘In conclusion, international scientific literature on influenza vaccination of teachers remains sparse and vaccination of this group is not nationally advised in most countries. However, Dutch media attention is growing, sparked by teacher shortages in recent years. The impact of vaccinating teachers has been particularly understudied, although some key experts and media reports have stated that any school board or other entity that is considering offering influenza vaccination to teachers needs to understand the impact of the vaccination. Our estimates showed that vaccinating teachers against influenza might be associated with a substantial decrease of sick leave days due to that viral infection. Teacher surveys, in-depth cost-effectiveness studies and a pilot at-school influenza vaccination programme could provide critical information about teacher vaccination.’ (page 25, lines 478-486)

6. PLOS authors have the option to publish the peer review history of their article (what does this mean?). If published, this will include your full peer review and any attached files.

Do you want your identity to be public for this peer review? For information about this choice, including consent withdrawal, please see our Privacy Policy.

Reviewer #1: No

Reviewer #2: Yes: Yoshiki Kusama

---

## [Decision Letter · Decision Letter 1]

30 May 2022

PONE-D-20-40234R1Influenza vaccination of school teachers: a scoping review and an impact estimationPLOS ONE

Dear Dr. Huiberts,

Thank you for submitting your manuscript to PLOS ONE. After careful consideration, we feel that it has merit but does not fully meet PLOS ONE’s publication criteria as it currently stands. Therefore, we invite you to submit a revised version of the manuscript that addresses the points raised during the review process.

We look forward to receiving your revised manuscript.

Kind regards,

Shinya Tsuzuki, MD, MSc

Academic Editor

PLOS ONE

Journal Requirements:

Additional Editor Comments :

Both reviewers favourably evaluated the manuscript but some minor concerns raised by them. Please respond to their comments before the final decision will be made.

Reviewers' comments:

Reviewer's Responses to Questions

**Comments to the Author**

1. If the authors have adequately addressed your comments raised in a previous round of review and you feel that this manuscript is now acceptable for publication, you may indicate that here to bypass the “Comments to the Author” section, enter your conflict of interest statement in the “Confidential to Editor” section, and submit your "Accept" recommendation.

Reviewer #2: All comments have been addressed

Reviewer #3: All comments have been addressed

2. Is the manuscript technically sound, and do the data support the conclusions?

Reviewer #2: Yes

Reviewer #3: Yes

3. Has the statistical analysis been performed appropriately and rigorously? 

Reviewer #2: Yes

Reviewer #3: Yes

4. Have the authors made all data underlying the findings in their manuscript fully available?

Reviewer #2: No

Reviewer #3: Yes

5. Is the manuscript presented in an intelligible fashion and written in standard English?

Reviewer #2: Yes

Reviewer #3: Yes

6. Review Comments to the Author

Reviewer #2: The authors have corrected their manuscript appropriately in accordance with my suggestions. The readability of the manuscript was dramatically improved by these revisions. However, the Discussion section is excessively long compared to the contents of the study. The following sentences are possibly omitted (or written more briefly). Please consider if the sentences should be removed or left.

Page 20, Lines 362–366

The other research…for teachers.

Page 21, Lines 380–384

Some key information…it are unknown.

Page 21, Lines 389–391

Future research…influenza vaccination.

Page 22, Lines 405–412

In 2019,…in the education sector.

Pages 22–23, Lines 421–423

These are relevant…on working days lost.”

Page 23, Lines 432–437

In school children…in older age.

I suggest that following sentences should be moved.

Pages 24–25, Lines 470–475

Our interview…other than nationally.

Page 22, Line 413 (because the sentences are not relevant to the impact estimation)

Reviewer #3: The authors have come up with an interesting way of bringing forth the impact of vaccination on teachers’ attendance in the Netherland. They have provided supporting data from various sources. This makes it worth publishing.

The authors have answered all the questions put forth by the earlier reviewers and made the necessary changes to the manuscript. They have edited the paper to see that the language used is grammatically correct. Also, they have changed the overall design by concentrating on the impact of influenza vaccination on the Netherlands’ teachers. The tables are very informative in understanding the calculations. The Result section provides all details. The supplementary data is in detail with relevant tables. The authors have put a lot of effort into this project and the information available is ample. They have correctly added the information associated with scoping review in the supplementary section.

The authors have put a lot of effort into correcting the language as well as reducing the content. Yet, the current version needs some more language polishing. Especially for better understanding. The most common issue I found was the sentences are too long. Breaking up the sentences will be helpful.

Below are a few examples of long sentences as well as grammatically incorrect one that needs to be rephrased into smaller sentences.

Line 110: some individual companies and organizations encourage their educational staff

111 to be vaccinated. For example, Amsterdam has offered free vaccination to schoolteachers since

112 2018/2019.

Line 112 rephrase

Line 114 “A review protocol was not registered for this study.”: rephrase

Line 203 “hits in dutch newspaper” rephrase

Line 257 “Absenteeism duration is measured yearly in nearly the total Dutch teacher population” : rephrase

Line 391-396: Long sentences

Line 405-407 Break up into two sentences

Line 419 However, at the perhaps earlier disease stage of sick leave for influenza, our results indicate that only 3.5-9.1 vaccinated teachers were needed to prevent 1 day of teacher influenza sick leave. Rewrite the sentence

Line 423 – 426 Too long

Line 428-431 Too long

Line 454 rephrase

Line 475 rephrase

7. PLOS authors have the option to publish the peer review history of their article (what does this mean?). If published, this will include your full peer review and any attached files.

Reviewer #2: **Yes: **Yoshiki Kusama

Reviewer #3: No

---

## [Author Response · Author response to Decision Letter 1]

10 Jul 2022

Dear PLOS ONE,

Thank you for your queries and for sending us the favourable evaluation of both reviewers. We gladly addressed the minor concerns raised by them as described below. We hope that we have adequately addressed their concerns. Feel free to contact us would there be any remaining queries. 

4. Have the authors made all data underlying the findings in their manuscript fully available?

Reviewer #2: No

Reviewer #3: Yes

The underlying data is indeed rather spread out across the manuscript. It is all available in multiple tables which can be found in the 5 supplements as follows: 

1. Scientific literature: table S1.2 

2. Grey literature: table S2.3

3. Interviews: table S3.1

4. Newspaper monitoring: table S4.2

5. Impact estimate: input parameters available in table 1 in the main manuscript and table S5.1, S5.2, and S5.3 in supplements 5.

6. Review Comments to the Author

Reviewer #2: The authors have corrected their manuscript appropriately in accordance with my suggestions. The readability of the manuscript was dramatically improved by these revisions. However, the Discussion section is excessively long compared to the contents of the study. The following sentences are possibly omitted (or written more briefly). Please consider if the sentences should be removed or left.

Page 20, Lines 362–366

The other research…for teachers.

Page 21, Lines 380–384

Some key information…it are unknown.

Page 21, Lines 389–391

Future research…influenza vaccination.

Page 22, Lines 405–412

In 2019,…in the education sector.

Pages 22–23, Lines 421–423

These are relevant…on working days lost.”

Page 23, Lines 432–437

In school children…in older age.

I suggest that following sentences should be moved.

Pages 24–25, Lines 470–475

Our interview…other than nationally.

Page 22, Line 413 (because the sentences are not relevant to the impact estimation)

Thank you. We now shortened the discussion by removing and shortening sentences. And we made some sections clearer by cutting sentences into two. In detail, the following changes were made:

- “The other research articles mainly assessed teachers’ attitudes towards influenza vaccination and the vaccination uptake in this group. However, most of the research was conducted in the USA, where influenza vaccination is already officially recommended for all citizens (excluding infants <6 months) and vaccination uptake is generally higher than in Europe. Grey literature showed that a few European countries nationally recommend influenza vaccination for teachers.” WAS CHANGED TO: “The other 11 papers mainly assessed teachers attitudes and uptake, mostly in the USA Grey literature showed that a few European countries nationally recommend influenza vaccination for teachers.” 

- “Some key informants viewed financial incentives as unnecessary or undesirable, but these incentives were reported in grey literature to increase vaccination uptake. Rotterdam decided not to offer free vaccination to teachers after consultation with school boards . Although vaccination uptake and its predictors have been studied in health care workers [50, 51], Dutch teachers’ knowledge of influenza vaccination and their attitudes regarding it are unknown.” WAS CHANGED TO: “Some key informants viewed financial incentives as unnecessary or undesirable, but these incentives were reported in grey literature to increase vaccination uptake. While vaccination uptake and its predictors have been studied in health care workers [50, 51], Dutch teachers’ knowledge of influenza vaccination and their attitudes regarding it are unknown.”

- “Future research should focus on estimating the effect of vaccinating teachers on sick leave, as many key informants expressed the need for understanding the potential impact of teacher influenza vaccination.” WAS CHANGED TO: “Many key informants expressed that they needed to know the potential impact of teacher influenza vaccination. Future research should also focus on estimating the effect on sick leave of vaccinating teachers.” 

- “In 2019, teacher shortages were 3.4%, 2.7% and 2.0% in the large cities of Amsterdam, Rotterdam and The Hague, respectively; in 2020 increases were expected in those cities to 8.4%, 6.6% and 4.7%, respectively [11]. In these three cities, requests have come from either the council’s education department (Amsterdam) or a political party (Rotterdam and The Hague) to set up a vaccination programme for school teachers or to investigate the feasibility of such a plan. While Dutch health care workers [51, 52] are offered free influenza vaccination to protect their vulnerable patients, decreased absenteeism and uninterrupted care are considered additional benefits [53]; this outcome could be similar in the education sector.” WAS CHANGED TO: “In 2019, teacher shortages were 3.4%, 2.7% and 2.0% in the large cities of Amsterdam, Rotterdam and The Hague, respectively. In 2020 these numbers were expected to reach 8.4%, 6.6% and 4.7%, respectively [11]. In these three cities, requests have come from either the council’s education department (Amsterdam) or a political party (Rotterdam and The Hague) to set up a vaccination programme for school teachers or to investigate the feasibility of such a plan. While Dutch health care workers [51, 52] are offered free influenza vaccination to protect their vulnerable patients, decreased absenteeism and uninterrupted care are considered additional benefits [53]. This outcome could be similar in the education sector for which potential uptake and predictors are not yet known.”

- “These are relevant results, as confirmed in a more recent Cochrane review from Demicheli (2018): “ We are uncertain about the effect of inactivated vaccines on working days lost.” WAS CHANGED TO: “These are relevant results, as there is uncertainty about the effect of vaccines on working days lost.”

- “In school children, influenza vaccination showed a positive effect on overall influenza morbidity, not only in schools [59] but also in the community [43, 60]. For the 2009 H1N1 influenza pandemic, it was found that vaccinating school children decreased absenteeism of both teachers and students [61]. Although we focused our research on short-term effects of influenza vaccination, it is important to note that a discussion is ongoing about the long-term effects of repeated vaccination at a young age, which might increase risk of influenza in older age [62].” WAS CHANGED TO: “In school children, influenza vaccination showed a positive effect on overall influenza morbidity, not only in schools [59] but also in the community [43, 60]. For the 2009 H1N1 influenza pandemic, it was found that vaccinating school children decreased absenteeism of both teachers and students [61].”

- “Our interviews found that Rotterdam school boards considered free influenza vaccination for their teachers, and some even implemented it. These school boards did not appear in our newspaper search (see S4), and therefore, it is to be expected that more school boards in the country offer their personnel influenza vaccination. Two foreign news articles (Australia and UK) also illustrate that, although teacher influenza vaccination is neither nationally advised nor offered for free, it is sometimes organized other than nationally [66-69] (see S1).” This was completely moved to supplement 3.

- The two sentences starting at line 413 are about how our impact estimation compares to the (little) available other research. Therefore we would prefer to leave it in the manuscript: ‘Although very limited data are available on the impact of influenza vaccination of specifically school teachers and staff on absenteeism, research has been reported on healthy adults in the general population. A Cochrane review by Demicheli 2014 found that in healthy adults, at least 71 people required vaccination to prevent one laboratory-confirmed case of influenza (CI: 64 to 80)[54].’ 

Reviewer #3: The authors have come up with an interesting way of bringing forth the impact of vaccination on teachers’ attendance in the Netherland. They have provided supporting data from various sources. This makes it worth publishing.

The authors have answered all the questions put forth by the earlier reviewers and made the necessary changes to the manuscript. They have edited the paper to see that the language used is grammatically correct. Also, they have changed the overall design by concentrating on the impact of influenza vaccination on the Netherlands’ teachers. The tables are very informative in understanding the calculations. The Result section provides all details. The supplementary data is in detail with relevant tables. The authors have put a lot of effort into this project and the information available is ample. They have correctly added the information associated with scoping review in the supplementary section.

The authors have put a lot of effort into correcting the language as well as reducing the content. Yet, the current version needs some more language polishing. Especially for better understanding. The most common issue I found was the sentences are too long. Breaking up the sentences will be helpful.

Below are a few examples of long sentences as well as grammatically incorrect one that needs to be rephrased into smaller sentences.

Line 110: some individual companies and organizations encourage their educational staff

111 to be vaccinated. For example, Amsterdam has offered free vaccination to schoolteachers since

112 2018/2019.

Line 112 rephrase

Line 114 “A review protocol was not registered for this study.”: rephrase

Line 203 “hits in dutch newspaper” rephrase

Line 257 “Absenteeism duration is measured yearly in nearly the total Dutch teacher population” : rephrase

Line 391-396: Long sentences

Line 405-407 Break up into two sentences

Line 419 However, at the perhaps earlier disease stage of sick leave for influenza, our results indicate that only 3.5-9.1 vaccinated teachers were needed to prevent 1 day of teacher influenza sick leave. Rewrite the sentence

Line 423 – 426 Too long

Line 428-431 Too long

Line 454 rephrase

Line 475 rephrase

Thank you. The full manuscript was revised by two of the authors. Long sentences were rewritten to make them more readable. Others were shortened. Most changes follow the reviewers above suggestions which were changed as follows:

- “In the absence of national guidance on influenza vaccination in the educational system, some individual companies and organisations encourage their staff to be vaccinated. For example, Amsterdam has offered free vaccination to schoolteachers since 2018/2019.” WAS CHANGED TO: “In the absence of national guidance on influenza vaccination in the educational system, some local companies and organisations encourage their staff to be vaccinated. For example, in Amsterdam, the public health service has offered free vaccination to schoolteachers since 2018/2019.”

- “A review protocol was not registered for this study.” This sentence is indeed redundant and out of context. As this information is available in supplement 6 we removed this sentence.

- “hits in Dutch newspapers” WAS CHANGED TO “the number of reports in Dutch newspapers”.

- “Absenteeism duration is measured yearly in nearly the total Dutch teacher population” WAS CHANGED TO: “Absenteeism duration is measured yearly in almost all Dutch teachers”.

- “For this, our impact calculations provide an initial estimate of sick leave due to cases of influenza infection that might be averted by vaccinating teacher population; the reduction in sick leave could be considerable with a moderate to high vaccination uptake. Future research should additionally focus on the cost-effectiveness of teacher vaccination both for schools and for public health generally, and it would ideally consider the potential effects of easier accessibility and financial incentives on the uptake.” WAS CHANGED TO: “Our impact calculations provide an initial estimate of influenza sick leave that might be averted by vaccinating the teacher population. The reduction in sick leave could be considerable with a moderate to high vaccination uptake. Future research should additionally focus on the cost-effectiveness of teacher vaccination both for schools and for public health generally. Ideally it would also consider the potential effects of easier accessibility and financial incentives on the uptake.”

- “In 2019, teacher shortages were 3.4%, 2.7% and 2.0% in the large cities of Amsterdam, Rotterdam and The Hague, respectively; in 2020 increases were expected in those cities to 8.4%, 6.6% and 4.7%, respectively [11]” WAS CHANGED TO: “In 2019, teacher shortages were 3.4%, 2.7% and 2.0% in the large cities of Amsterdam, Rotterdam and The Hague, respectively. In 2020 these numbers were expected to reach 8.4%, 6.6% and 4.7%, respectively [11]”

- “However, at the perhaps earlier disease stage of sick leave for influenza, our results indicate that only 3.5-9.1 vaccinated teachers were needed to prevent 1 day of teacher influenza sick leave.” WAS CHANGED TO: “Laboratory testing and physician consultation might occur relatively late in the disease process. At the earlier disease stage of taking sick leave for influenza, our results indicate that only 3.5-9.1 vaccinated teachers were needed to prevent 1 day of teacher influenza sick leave.”

- “Our relatively high estimates of potentially averted absenteeism in teachers may be likely, given the conclusion of Nichol et al (1995) that immunization decreased absenteeism from work due to upper respiratory illness by 43% and absenteeism due to all illnesses by 36% in healthy, working adults [56].” WAS CHANGED TO: “Our relatively high estimates of potentially averted absenteeism in teachers may be likely, given the conclusion of Nichol et al (1995). They reported that immunization decreased absenteeism from work due to upper respiratory illness by 43% and absenteeism due to all illnesses by 36% in healthy, working adults [56].” 

- “Also, a review study of health care workers concluded absenteeism due to influenza-like-illness was reduced and its duration shortened in those vaccinated, but an estimate of the magnitude of reduction was unavailable [58].” WAS CHANGED TO: “Another review study of health care workers concluded that absenteeism was less frequent and shorter in those vaccinated, but the magnitude of this reduction was unavailable [58]. 

- “As some teachers were eligible for influenza vaccination because of age (60+) or underlying medical conditions, we expect that we have underestimated, rather than overestimated, impact of vaccinating teachers.” WAS CHANGED TO: “But, some teachers probably had received vaccination from their GP because of age (60+) or underlying medical conditions (within the general immunization programme). The actually registered sick leave that we used as input parameter would have been higher had they not been vaccinated within the general immunization programme. Therefore, we expect that we have underestimated, rather than overestimated, impact of vaccinating teachers. 

- “Two foreign news articles (Australia and UK) also illustrate that, although teacher influenza vaccination is neither nationally advised nor offered for free, it is sometimes organized other than nationally [66-69].” WAS CHANGED TO: “Two foreign news articles (Australia and UK) also illustrate that, although teacher influenza vaccination is neither advised nor offered for free at a national scale, it is sometimes organized locally [66-69]”. This paragraph was moved to supplement 3.

7. PLOS authors have the option to publish the peer review history of their article (what does this mean?). If published, this will include your full peer review and any attached files.

Do you want your identity to be public for this peer review? For information about this choice, including consent withdrawal, please see our Privacy Policy.

Reviewer #2: Yes: Yoshiki Kusama

Reviewer #3: No

---

## [Editor Report · Decision Letter 2]

19 Jul 2022

Influenza vaccination of school teachers: a scoping review and an impact estimation

PONE-D-20-40234R2

Dear Dr. Huiberts,

We’re pleased to inform you that your manuscript has been judged scientifically suitable for publication and will be formally accepted for publication once it meets all outstanding technical requirements.

Kind regards,

Shinya Tsuzuki, MD, MSc

Academic Editor

PLOS ONE

Additional Editor Comments (optional):

All comments raised by reviewers were now appropriately answered.
---

## [Editor Report · Acceptance letter]

2 Aug 2022

PONE-D-20-40234R2 

Influenza vaccination of school teachers: a scoping review and an impact estimation 

Dear Dr. Huiberts:

I'm pleased to inform you that your manuscript has been deemed suitable for publication in PLOS ONE. Congratulations! Your manuscript is now with our production department. 

Kind regards, 

on behalf of

Dr. Shinya Tsuzuki 

Academic Editor

PLOS ONE